# TrkA Co-Receptors: The Janus Face of TrkA?

**DOI:** 10.3390/cancers15071943

**Published:** 2023-03-23

**Authors:** Sarah Trouvilliez, Chann Lagadec, Robert-Alain Toillon

**Affiliations:** 1Univ. Lille, CNRS, INSERM, CHU Lille, UMR9020-U1277-CANTHER-Cancer Heterogeneity Plasticity and Resistance to Therapies, OncoLille Institute, Bvd. du Professeur Jules Leclercq, F-59000 Lille, France; 2GdR2082 APPICOM-«Approche Intégrative Pour Une Compréhension Multi-Échelles de la Fonction des Protéines Membranaires», 75016 Paris, France

**Keywords:** TrkA, co-receptors, aggressiveness, drug resistance

## Abstract

**Simple Summary:**

NGF was the first growth factor discovered by Rita Levi Montalcini in 1950. TrkA, its high affinity receptor, is an oncogene that is overexpressed in many cancers. However, targeted therapies against TrkA, in particular kinase inhibitors, have not yet demonstrated efficacy in the context of overexpression. In this review, after describing the state-of-the-art TrkA-targeted therapies, we will elicit the failures of these therapies by focusing on non-genomic resistance.

**Abstract:**

Larotrectinib and Entrectinib are specific pan-Trk tyrosine kinase inhibitors (TKIs) approved by the Food and Drug Administration (FDA) in 2018 for cancers with an NTRK fusion. Despite initial enthusiasm for these compounds, the French agency (HAS) recently reported their lack of efficacy. In addition, primary and secondary resistance to these TKIs has been observed in the absence of other mutations in cancers with an NTRK fusion. Furthermore, when TrkA is overexpressed, it promotes ligand-independent activation, bypassing the TKI. All of these clinical and experimental observations show that genetics does not explain all therapeutic failures. It is therefore necessary to explore new hypotheses to explain these failures. This review summarizes the current status of therapeutic strategies with TrkA inhibitors, focusing on the mechanisms potentially involved in these failures and more specifically on the role of TrkA.

## 1. Introduction

The tyrosine kinase receptor TrkA, the high-affinity receptor for Nerve Growth Factor (NGF), is essential for both the survival and differentiation of neural cells. TrkA, encoded by the NTRK1 gene, was first discovered as a fusion oncogene in colon cancer by Martin-Zanca et al. [1]. Subsequently, this oncogene was detected in other cancers, such as human breast tumor cells and papillary thyroid carcinoma [2,3]. Somatic rearrangements occur between the tyrosine kinase domain of TrkA and the 5′ end fused to the gene encoding another protein (TPM3, TP53, ATP1B, etc.). The result of this genomic event is a constitutively activated chimeric protein. In addition, point mutations, missense mutations and deletions of TrkA have been described in congenital insensitivity to pain and cancer. These mutations can affect the three parts of TrkA: extracellular, transmembrane and intracellular. In cancer, mutations have been described that increase TrkA activation independently of ligand binding or through the modulation of ligand affinity binding. For example, the P203A mutation in the extracellular domain increases NGF binding to TrkA, and deleted ΔTrkA (75 AA of the extracellular domain) and NTRK fusions result in its constitutive activation [4,5,6]. In recent years, special attention has been paid to NTRK1 fusions in cancers. Indeed, in several cancers, cells with NTRK1 fusions have been shown to be responsive to tyrosine kinase inhibitors (TKIs) in vitro and in vivo. For this reason, several molecules have been designed including belizatinib, AZ23, Cpd5n, PHA-E429, milciclib, GNF-5837, cabozantinib, sitravatinib, altiratinib, Larotrectinib or Entrectinib. Nowadays, two inhibitors, Larotrectinib and Entrectinib have been accepted by various drug agencies in the world. Larotrectinib (Vitrakvi^®^) is a specific pan-Trk inhibitor, which was approved by the Food and Drug Administration (FDA) in November 2018 for cancers with an NTRK fusion. The approval of Larotrectinib was based on three clinical trials: LOXO-TRK-14001 (NCT02122913), SCOUT (NCT02637687), and NAVIGATE (NCT02576431). In addition, Entrectinib (Rozlytrek^®^) was agreed on by the FDA about a year later (August 2019) for patients with NTRK fusions. The development of this drug was a real breakthrough in cancer treatment. In fact, unlike the traditional cancer strategy treatment, Larotrectinib and Entrectinib are approved based on biomarkers (NTRK fusion) independent of the tumor cell type. Other countries have quickly approved these drugs with different restrictions. In France, Larotrectinib received temporary authorization for the cohort between April and November 2019 and then a conditional European visa in September 2019. However, only one year later (July 2020), the French drug agency considered the inputs of Larotrectinib as moderate for infantile fibrosarcoma and soft tissue sarcoma with NTRK fusions and as insufficient for the other cancers. These disappointing conclusions were reinforced by several clinical trials in which patient responses were absent or poor. To explain Trk inhibitor failure, point mutations that abolish inhibitor binding have been proposed. To overcome this resistance mechanism, new Trk inhibitors with different binding modes have been generated: XL184 MGCD516, DS-6051, TSR-011, DCC2701, LOXO-195 and TPX-0005. However, primary and secondary TKI resistance has been observed in the absence of mutations in cancers with NTRK fusions, which challenges these hypotheses. Moreover, when TrkA is overexpressed, despite the fact that it promotes TrkA ligand-independent activation, TKIs are not efficient. It is apparent that genetic factors cannot explain all the mechanisms involved and it is necessary to investigate other possibilities to explain these therapeutic failures.

In this review, the therapeutic potential of NGF/TrkA inhibitors (anti-NGF and Trk TKIs) in cancers as curative and palliative drugs will be presented. Then, the mechanisms involved in pan-Trk inhibitor failure will be analyzed and, in particular, besides the “genetic” mechanisms, we will highlight the role of TrkA co-receptors.

## 2. State of the Art: Therapeutic Potential of the NGF/TrkA Axis

Many studies have reported the importance of the NGF/TrkA axis in cancers. As a result, molecules that block NGF binding to it receptors (TrkA and P75^NTR^) or Trk phosphorylation have been generated (Figure 1). Here, we discuss the therapeutic potential of these 2 types of compounds for the treatment of cancer in a curative or palliative way.

### 2.1. Anti-(Pro)NGF: The Next Generation of Anti-Pain?

The NGF/TrkA axis has been targeted with Tanezumab, an antibody against NGF. The efficacy of this molecule in the treatment of pain was first demonstrated in osteoarthritis of the knee and hip and chronic low back pain [7,8]. In addition, in cancer, a recent phase III study showed that Tanezumab reduces pain caused by bone metastases [9]. However, Tanezumab has been associated with adverse effects, including arthralgia and paresthesia [8]. This is probably due to the fact that (pro)NGF can also bind P75^NTR^ [10,11]. P75^NTR^ is the low affinity neurotrophin receptor. It belongs to the family of death receptors. In its extracellular part, it presents cysteine-rich domains (CRD) characteristic of the tumor necrosis factor receptor (TNFR) family [12], but also intracellular death domains [13]. P75^NTR^ is thus able to bind NGF but also all neurotrophins and their precursors (such as proNGF). P75^NTR^ has long been described as enhancing the action of neurotophins and NGF on their receptors, such as TrkA. In a recent review, Conroy and Coulson elegantly describe the complexity of the interaction between TrkA and p75^NTR^ from a model of ligand passing model switching to the current theory of allosteric regulation of TrkA by p75^NTR^ [14]. It is also known that, under NGF stimulation, p75^NTR^ can act through its own signaling pathways and notably in the mediation of pain [15].

Nevertheless, to avoid these side effects, anti-NGF inhibitors with different binding modes than those of tanezumab have been developed, such as peptide A2 or BVNP-0197 (peptide that binds loop II/IV of NGF) [16,17].

### 2.2. Kinase Inhibitors: The Next Generation of Targeted Therapies?

Since TrkA is a receptor with tyrosine kinase activity, tyrosine kinase inhibitors have been developed to block TrkA activation associated with an NTRK1 fusion or TrkA overexpression in cancers [18]. As with many TKIs, the pan-Trk inhibitors Lestaurtinib, Larotrectinib and Entrectinib bind to the kinase pocket of the receptor to prevent ATP binding (ATP competitor) and subsequent phosphorylation.

Lestaurtinib (CEP-701) is an oral analog of K252a. These molecules are indozolocarbazole staurosporine-derivative inhibitors. It was the first TrkA inhibitor to show activity in vitro and in vivo (preclinical model) in prostate cancer [19]. Other studies suggest that K252a also has an effect on pancreatic and breast cancer [20,21]. However, the results of clinical trials in both solid and blood cancers have been disappointing. For example, Lestaurtinib failed to achieve the primary endpoint of reducing circulating prostate specific antigen (PSA) levels in the blood [22]. It has also been suggested that the bioavailability of CEP-701 is not optimal and does not allow for achieving the desired therapeutic effect [22]. In pancreatic cancer, a combination of gemcitabine and CEP-701 was evaluated (Phase I clinical trial). This trial was discontinued due to a lack of benefit and side effects (nausea, vomiting, fever, etc.) [23]. The side effects of CEP-701 were attributed to its lack of specificity. In fact, CEP-701 is a multi-kinase inhibitor, inhibiting, in addition to TrkA, TrkB, TrkC, FLT3, FGFR and JAK2. More recently, an intrinsic resistance mechanism involving NF-kappa B signaling has been described for CEP-701 [24].

To overcome the low bioavailability and specificity of Lestaurtinib, new generations of Trk inhibitors have been designed. Two of them are approved by the FDA (Food and Drug Association): Larotrectinib (LOXO-101) and Entrectinib. Both Larotrectinib and Entrectinib inhibit the phosphorylation of the three members of the Trk family (TrkA, B and C), but Larotrectinib is more specific than Entrectinib, as it also inhibits ALK and ROS. Interestingly, unlike Lestaurtinib, Larotrectinib and Entrectinib have been approved only for cancers with an oncogenic NTRK1 fusion [25]. NTRK1 fusions result in the overexpression of a chimeric protein lacking extracellular and transmembrane domains of TrkA. For example, the TPM3–NTRK1 fusion protein consists of the N-terminal domain of TPM3 (codons 1–221) and the transmembrane/intracellular domains of TrkA (codons 419–end). Fusion proteins with intracellular (cytoplasmic) localization exhibit ligand-independent activation. As a result, TrkA signaling pathways (MAPK, Akt, PLC) are sequentially activated. This oncogene dependency explains why cancers harboring fusions show a primary response to Larotrectinib and/or Entrectinib [4]. A clinical trial on 159 adult and pediatric patients with TRK fusion-positive solid tumors showed that treatment with Larotrectinib resulted in a median response rate of 35.2 months and a median progression-free survival of 28.3 months [26]. In addition, Larotrectinib may be more effective in improving quality of life with pediatric and adult cancers than Entrectinib [27]. Larotrectinib also appears to be more effective than Entrectinib for non-small cell lung cancer. In fact, treatment with Larotrectinib and Entrectinib resulted in 5.4 and 1.2 median life years with progression and 7.0 and 1.8 median overall life years, respectively [28].

### 2.3. Second-Generation Trk Inhibitors for Cancers with Point Mutation(s)

Indeed, point mutations have been reported to cause secondary resistance to Larotrectinib and Entrectinib. These point mutations affect different residues close to the TrkA ATP-binding pocket: F589L, G595R, G667C, G667S, V573M. These point mutations have also been reported to be responsible for primary resistance [29]. As a result, TKI fixation to the ATPase site of TrkA is impaired. To overcome this resistance, LOXO-195 was designed to bind TrkA differently than LOXO-101 [30].

## 3. Genomic Alterations Cannot Recapitulate All Resistance Mechanisms

While there is great hope for Larotrectinib and Entrectinib, it has become apparent that the contribution of these molecules, in particular Larotectinib, seems to be moderate and even insufficient (HAS, technical control, 22 September 2021). It is therefore questionable whether the elements of such a “failure” were already present in the previous results with CEP-701. Moreover, the fact that Larotrectinib (or Entrectinib) is only effective for oncogenic fusions of TrkA and not the overexpression of TrkA suggests that the oncogenic activity of TrkA is not only related to its kinase activity.

### 3.1. Is TrkA TKI Resistance Related to Co-Alterations of Other Oncogenic Pathways?

TrkA is expressed in normal breast epithelial cells, and ligand fixation leads to TrkA phosphorylation, as in cancer cells. However, this phosphorylation does not promote proliferation and survival as in breast cancer cell lines [31]. This is probably due to the fact that in cancer cells, downregulation of the phosphatase may lead to less negative feedback from the kinase [32]. This observation highlights that the control of TrkA signaling and its cellular consequence depend not only on it conformation change and it phosphorylation but on other molecular actors. As demonstrated for CEP-701 and NF-kappa B pathway activation, one may wonder whether intracellular signaling can explain resistance to TrkA. Indeed, co-alterations of TrkA often affect TrkA signaling molecules: PI3K signaling (61% of patient samples) and MAPK pathways (32% of patient samples). In addition, these mutations can also alter the cell cycle machinery (58% of patient samples) and tyrosine kinase families (58% of patient samples) [29]. Thus, it has been directly demonstrated that overexpression of the MAP-kinase pathway in cancers harboring an NTRK1 fusion is responsible for resistance to Larotrectinib and LOXO-195 [33].

### 3.2. Can TrkA and Their Co-Receptors Elicit TKI Lack of Efficiency?

The cut-off that defines an NTRK fusion-positive tumor is 15% of cells. Thus, a large number of cells of the NTRK fusion-positive tumor do not express the chimeric Trk protein, but rather a full length TrkA (including its extracellular portion). The effect of TKIs on cancer cells overexpressing TrkA has been less studied compared to that in cells harboring an NTRK fusion. Using MCF10A immortalized breast cells, Kyker-Snowman and colleagues tested the effect of Larotrectinib on non-mutated TrkA-overexpressing cells. Interestingly, they showed that Larotrectinib inhibited TrkA phosphorylation and cell survival in vitro [34]. However, the in vivo experiment was performed with the breast cancer cell line MCF7, which overexpresses TrkA, and not with the “normal” breast cell line MCF10A. Thus, this work does not allow for conclusions on the potential of Larotrectinib in vivo when TrkA is overexpressed. Moreover, to the best of our knowledge, Larotrectinib and Entrectinib have not demonstrated an effect during a clinical trial in cancers overexpressing TrkA, similar to Lestaurtinib. Because the primary function of the extracellular and transmembrane domains of TrkA is ligand binding and conformational change, it has been hypothesized that this resistance results from TrkA homodimerization and transactivation induced by NGF and/or ProNGF. However, molecules blocking ligand fixation on TrkA are not able to abolish all TrkA pro-tumorigenic effects. Therefore, TrkA co-receptor(s) may modulate intracellular molecules recruited and their activation. Indeed, numerous studies have focused on the P75^NTR^/TrkA interaction to explain such discrepancies. It appears that the TrkA interactome is much more complex. Thus, we will discuss how TrkA/co-receptor association modulates TrkA structure/function and may contribute to TKI failure through the (1) modulation of TrkA ligand affinity, (2) TrkA phosphorylation, or (3) TrkA phospho-independent pathways.

#### 3.2.1. TrkA Heterocomplexes May Favor Ligand Binding

The formation of the P75^NTR^/TrkA heterodimer (stoichiometry 1:2:1) induces a 100-fold increase in NGF affinity for TrkA [35,36]. As we have previously described, two mechanisms have been proposed to explain how P75^NTR^ increases NGF affinity for TrkA via its extracellular domain [14]. The most likely model is the ligand passing model: P75^NTR^ presents NGF to TrkA and thus does not interact directly with TrkA. This indirect extracellular interaction of P75^NTR^/TrkA promotes TrkA activation, signaling and thus the pro-tumorigenic effects of TrkA.

In addition, we have recently shown that CD44 may interact with TrkA [37]. The glycoprotein CD44 (cluster of differentiation 44) is encoded by the CD44 gene (20 exons) located on locus 11p13. The transcription of CD44 allows for the production of a standard form consisting of exons 1 to 5 and exons 6 to 10. Alternative splicing of the 10 central exons of CD44 allows for the production of more than 20 isoforms [38]. Two of these isoforms, CD44v3 and CD44v6, have been described as promoting ligand binding to RTKs [39,40,41]. In particular, EGF binding to the heparan sulfate chain of CD44v3 promotes its binding to EGFR and its activation. Interestingly, we previously demonstrated that the TrkA/CD44v3 interaction involves the C-ter region of CD44v3 to LRR1 of TrkA [42]. Hence, the deglycosylated LRR1 has been shown to have a higher affinity for NGF. We hypothesized that CD44v3 may potentiate NGF binding through its interaction with LRR1 (Figure 2).

#### 3.2.2. TrkA Heterocomplexes Regulate Its Phosphorylation

Co-receptors can also regulate TrkA activation via their intracellular parts (Figure 3).

The TrkA/P75^NTR^ association does not only involve the extracellular parts of these two receptors. In fact, the formation of the TrkA/P75^NTR^ complex has been proposed to rely on intracellular interactions with the receptor Kidins220 [43]. The intracellular parts of the receptors are also critical for the pro-tumor effect of this complex. First, P75^NTR^ delays TrkA ubiquitination by TRAF6 and thus TrkA internalization [44,45]. NGF stimulation of TrkA also promotes P75^NTR^ cleavage, and the intracellular fragment of P75^NTR^, termed P75^NTR^-ICD, induces a conformational change in the extracellular domain of TrkA that promotes its signaling [46]. In turn, TrkA activation increases P75^NTR^ cleavage by ADAM17, which produces the P75^NTR^-CTF fragment. This fragment is relocalized to endosomes where it is cleaved by gamma-secretase to produce the P75^NTR^-ICD fragment [47]. In addition, G-protein coupled receptors (GPCRs) transactivate TrkA. In PC12 cells, the A2a receptor, after activation by adenyl cyclase, activates TrkA, which promotes Akt, ERK1/2 and SAPK/JNK, but not p38 MAPK. These mechanisms are calcium-dependent [48,49]. In addition, other GPCR ligands have been reported to support TrkA transactivation in PC-12 cells, including PACAP (pituitary adenylate cyclase-activating protein) or LPA1 (lysophosphatidate 1). In NG108 cells, activation of the AT2 angiotensin 2 receptor by Ang II induces the phosphorylation of TrkA. TrkA activation results in sustained MAPK (p42/p44) activation that promotes neurite outgrowth [50]. The MAPK activation mechanism does not rely on PKA (protein kinase A) activation, but on Fyn, a member of the Src family of kinases [51]. In addition, GPCR activation can be ligand-independent. For example, NOX-2 (NADPH oxidase 2) can activate the GPCR FPR1 via p47phox phosphorylation and thus TrkA transactivation [52].

Members of the epidermal growth factor family have also been shown to be involved in the control of TrkA activation. Cross-talk between TrkA and EGFR signaling has been demonstrated in monocytes. Indeed, the inhibition of TrkA phosphorylation results in EGFR inhibition, and the specific EGFR inhibitor tyrphostin AG1478 abolishes TrkA phosphorylation. Furthermore, the knockdown of EGFR by siRNA abolishes NGF-induced TrkA phosphorylation (tyrosine 490) and reciprocal siTrkA abolishes EGF-induced EGFR phosphorylation [53]. More recently, in cancer, the combination of the TrkA inhibitor GNF5837 (ATP competitor) and the EGFR specific inhibitor erlotinib was demonstrated to have efficiency in vitro and in vivo [54]. In addition, the HER2 receptor (p185HER2) is overexpressed in breast cancer (30%) and is associated with reduced free survival and brain metastasis [55]. Interestingly, Tagliabue et al. have shown, via coimmunoprecipitation, that NGF stimulation promotes TrkA/HER2 complex formation, which is followed by HER2 phosphorylation, activating MAPK signaling and thus promoting cell proliferation [56].

#### 3.2.3. TrkA Heterocomplexes Permit the Activation of a Phospho-Independent Pathway

We have shown that CD44v3 interaction with TrkA is associated with a phospho-independent signaling pathway involving p115RhoGEF/RhoA/RhoC/ROCK1 [37,42]. Interestingly, this signaling is not affected by K252a and independent of TrkA phosphorylation (as demonstrated by a kinase-dead mutation). P75^NTR^ can also activate GTPases, including RhoA and NGF, leading to its inactivation [57,58,59]. Thus, dynamic RhoA activation allows for the dynamic activation of actin and thus migration (cancer cells) or differentiation (neurons). Further studies are needed to understand the role of TrkA/P75^NTR^ and NGF/TrkA/CD44v3 in dynamic RhoA activation.

TrkA phosphorylation can also be induced by ProNGF, which promotes other resistance mechanism(s). Recently, we have demonstrated the role of EphA2 (Ephrin type A receptor 2) in breast cancer aggressiveness and TrkA TKI resistance [60]. EphA2 is a 135 kDa tyrosine kinase receptor. This receptor belongs to the A family of Eph kinases. EphA2 has eight different ligands (ephrin A) anchored to the cell membrane [61]. Its overexpression has been reported in various cancers, including breast cancer, and is associated with higher tumor aggressiveness. In breast cancer, EphA2 overexpression is associated with the most aggressive triple-negative breast cancer subtype [62]. Overexpression of the EphA2 receptor in MCF10A cells leads to their malignant transformation. Overexpression causes defects in cell–cell contacts [63]. In addition, low-molecular weight tyrosine phosphatase has been reported to inhibit EphA2 phosphorylation [63]. However, EphA2 can be phosphorylated via a ligand-independent mechanism. The phosphorylation of serine 897 of EphA2 by Akt promotes the relocalization of EphA2 to the leading edge, thereby promoting migration induced by various growth factors [64]. In contrast, ligand fixation on EphA2 inhibits EphA2 relocalization and chemotaxis. In addition, our team has identified another mechanism by which EphA2 promotes metastasis via ProNGF. Indeed, in breast cancer, ProNGF stimulation leads to the formation of the TrkA/EphA2/Sortilin complex, which induces the phosphorylation of Src through a pTrkA-independent mechanism. Interestingly, using siRNA we highlight that this signaling is dependent on EphA2 [65]. Thus, tyrosine kinase receptors could, independently of their kinase, make use of the functionalities of other membrane proteins to induce pro-oncogenic signaling, as recently suggested by Thomas and Weihua for EGFR [66]. Targeting TrkA/CD44v3 and TrkA/EphA2 and their downstream signaling pathways may be of interest for tumors with high NGF and ProNGF staining, respectively (Figure 4).

## 4. What Could Be Done to Counteract the Oncogenic Effects of TrkA?

In this review, we have highlighted the role of (pro)NGF/TrkA inhibitors in cancer therapeutic strategies. In addition, we explore non-genomic mechanisms that contribute to TKI therapeutic failure: Trk homodimerization and co-receptors. We described three mechanisms of action of TrkA co-receptors: ligand affinity enhancement, TrkA transactivation, activation of phospho-independent pathway. In this context, is it possible to propose therapeutic solutions to counteract the oncogenic effects of TrkA?

TKI can be first used in combination with protein–protein inhibitors (PPI). The choice of PPI can be made according to the TrkA/co-receptor complex(s) harbored by the tumor. However, this approach requires the investigation of optimal concentrations of the two compounds. In addition, as a TKI, PPI inhibitors may bind not only to TrkA but also to TrkB/C and cause side effects. To avoid this problem, allosteric TrkA inhibitors (type III) have been developed that bind specifically to the juxtamembrane region of TrkA [67,68,69]. The design of a new allosteric TrkA inhibitor that inhibits TrkA/CD44v3 signaling may be a new therapeutic opportunity to prevent metastasis in triple negative breast cancer.

However, this approach may lead to secondary resistance due to the interaction of TrkA with other co-receptors. Therefore, TrkA inhibitors that act as molecular glue to stabilize TrkA dimers have been developed (VMD-928). These inhibitors are selective for TrkA and bind to the ATP-binding site [70]. The efficacy of VMD-928 is currently being evaluated in patients with various cancers that overexpress TrkA (adenoid cystic carcinoma, cholangiocarcinoma, lung cancer, pancreatic cancer, parotid gland cancer and squamous cell carcinoma of the head and neck; Phase I clinical trial) [71].

To avoid side effects and ensure efficacy, ADC (Antibody Drug Conjugated) has been developed in recent years. ADCs are composed of a humanized antibody that recognizes membrane proteins specific to cancer cells, a linker, and a payload that is highly cytotoxic (tubulin inhibitors, DNA-damaging agents) [72]. One can thus imagine a strategy to recognize TrkA complexes (TrkA/EphA2 or TrkA/CD44v3) with a bivalent antibody of an ADC type.

## 5. Conclusions

To achieve the desired efficacy for TrkA inhibitors, precise knowledge of TrkA biology and its regulations is definitely required. However, understanding the mechanisms that regulate Trk activation could have implications not only for cancer treatment but also for the treatment of other diseases, including chronic pain, inflammatory diseases or neuropathology.

## Figures and Tables

**Figure 1 cancers-15-01943-f001:**
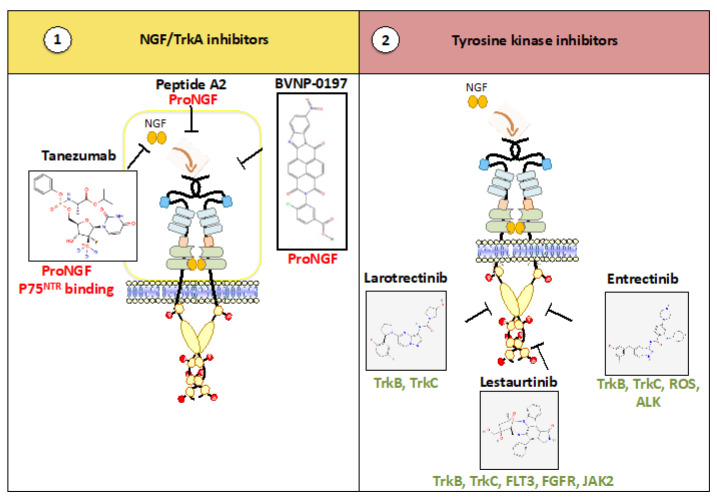
NGF/TrkA inhibitors. Two main types of molecules have been designed to impede NGF/TrkA signaling (**1**,**2**). First are molecules that block the NGF/TrkA association (**1**). Tanezumab is a monoclonal antibody that binds NGF. Peptide A2 and BVNP-0197 also bind NGF but do not promote P75^NTR^ signaling. Second are compounds that inhibit TrkA phosphorylation (**2**). The three inhibitors are ATP competitors but have different specificities and inhibit other receptors (green).

**Figure 2 cancers-15-01943-f002:**
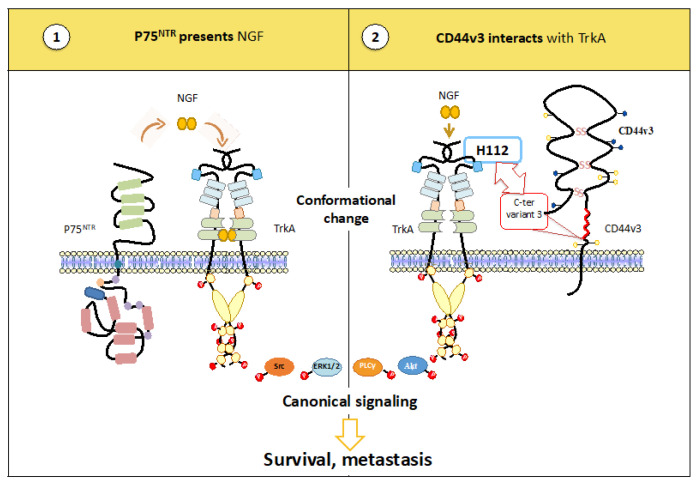
Comparison of the role of P75^NTR^ and CD44v3 in NGF binding to TrkA. P75^NTR^ presents NGF to TrkA (**1**), whereas CD44v3 interacts with TrkA in the LRR1 NGF-binding region (**2**).

**Figure 3 cancers-15-01943-f003:**
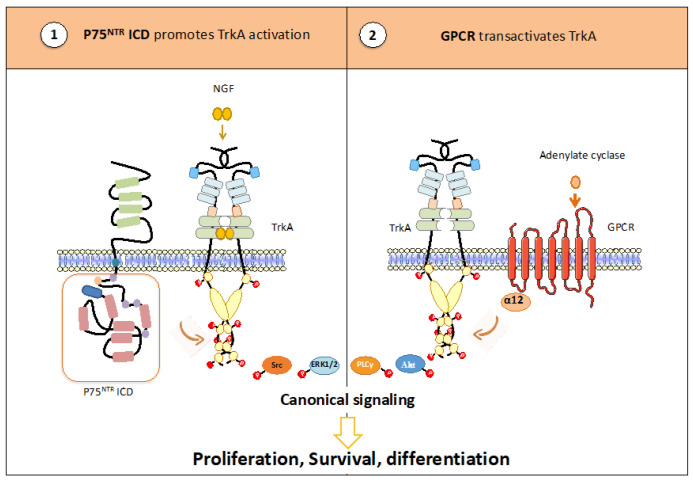
P75^NTR^ (**1**) and GPCR (**2**) promote the activation of TrkA via their intracellular parts. Following NGF stimulation, the intracellular part of P75^NTR^ is cleaved and promotes TrkA phosphorylation. In addition, GPCR transactivates TrkA in a ligand-independent-manner.

**Figure 4 cancers-15-01943-f004:**
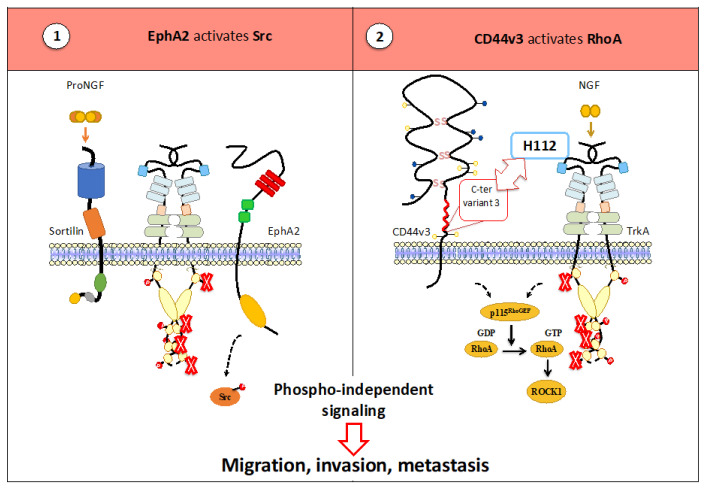
EphA2 and CD44v3 promote the activation of a TrkA phospho-independent pathway. (**1**) Following ProNGF stimulation, the TrkA/EphA2 association occurs and permits Src phosphorylation independently of TrkA phosphorylation. (**2**) Following NGF stimulation, TrkA interacts with CD44v3. CD44v3 permits RhoA activation independently of TrkA phosphorylation. As a result, TrkA/CD44v3 and TrkA/EphA2 contribute to breast cancer aggressiveness and TKI failure.

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
