# Peer review of "TrkA Co-Receptors: The Janus Face of TrkA?"

_cancers, 2023, doi:10.3390/cancers15071943_

Round 1

Reviewer 1 Report

The review from Trouvilliez et al., is focused on the discussion of possible mechanisms underlying failure of current FDA approved cancers therapy targeting TrkA pathway through tyrosine kinase inhibitors.

The topic is certainly of interest for the researchers of the field as well as for clinicians. However, its readability and Figures must be improved.

-Fig.1 Please add symbols indicating the block or the activation of the pathway.

-Fig.2 Please, include arguments and /or references about the proposed involvement of NGF in the CD44-TrkA pathway, as the authors state: "and potentially NGF binding..").

Figs 3-4: is aggressiveness the only outcomes/readouts of the proposed signalling?

-Par 2.1: Please briefly introduce ProNGF-p75NTR signalling pathway and explain why and how ProNGF (in the title) is of relevance for the p75NTR dependent anti-pain therapy.

Overall, the review is quite difficult to follow: entire paragraphs appears a plain list of published findings lacking a coherent and systematic perspective. A partial re-elaboration of the manuscript-at least from par 3.1- presenting the current literature in a fluent and argumented writing would definitively be desirable. The perspective paragraph is far too long, it should be the core of the manuscript when revised as suggested above.

Reviewer 2 Report

This manuscript presents a thorough summary of current knowledge on the role of TrkA mutations in human tumorigenesis, the significance of TrkA-targeting therapies, and genomic and non-genomic mechanisms of primary and secondary resistance against such therapeutical approaches. The main focus of the review is on the signaling partners of TrkA (termed co-receptors in the paper, e.g. p75NTR, CD44v3, GPCRs, Epha2) and their possible role in the development of resistance to TrkA-specific tyrosine kinase inhibitors. The manuscript compiles useful information on this important topic. However, the text contains typing, grammatical, and even factual errors (especially in the paragraph on page 7-8, see below) that need to be corrected.

The list of errors:

   line 48: ...inhibitors, Larotrectinib and Entrectinib...                         

   line 49:  ...pan-Trk, TKI...

   line 175:  ...to impede...

   line 176: ...They have different off target (what does that mean?)

   line 177:  ...that binds to NGF...

   line 179:  ...ATP competitors...

   line 282:  ...heterotrimeric G-proteins;  G-proteins are inactive (GDP-bound) or active (GTP-bound)

   line 284:  ...free protein G ...  does this refer to the free alpha-subunit of the G-protein?  Activation is exerted by the active, GTP-bound form of the alpha-subunit (and not by GTP hydrolysis.)

   line 296:  NOX-2 is an enzyme, not a reactive oxygen species.

   line 304:  Gene silencing by siRNA is called knockdown (not knockout)

   Figure 3/2:  AT2. Adenylate cyclase does not belong there, should be erased.

   line 323:  ...Interestingly, etc...  This sentence is confusing, should be rewritten.

   Table 1:  Contains many typos, even in the title  (Cancer cells specific, Combinaison, Theorical).

Author Response

We would like to thank reviewer 2 for his corrections. We have modified the text to take them into account. Table 1 has been removed to reduce the size of the manuscript.

Round 2

Reviewer 1 Report

The review has really improved, overcoming some initial readability issues. The figures are now clear, and the paragraphs structure is good.